# Identifying gene targets for brain-related traits using transcriptomic and methylomic data from blood

Ting Qi et al.#

Understanding the difference in genetic regulation of gene expression between brain and blood is important for discovering genes for brain-related traits and disorders. Here, we estimate the correlation of genetic effects at the top-associated *cis*-expression or -DNA methylation (DNAm) quantitative trait loci (*cis*-eQTLs or *cis*-mQTLs) between brain and blood ($r_b$). Using publicly available data, we find that genetic effects at the top *cis*-eQTLs or mQTLs are highly correlated between independent brain and blood samples ($\hat{r}_b = 0.70$ for *cis*-eQTLs and $\hat{r}_b = 0.78$ for *cis*-mQTLs). Using meta-analyzed brain *cis*-eQTL/mQTL data ($n = 526$ to $1194$), we identify 61 genes and 167 DNAm sites associated with four brain-related phenotypes, most of which are a subset of the discoveries (97 genes and 295 DNAm sites) using data from blood with larger sample sizes ($n = 1980$ to $14,115$). Our results demonstrate the gain of power in gene discovery for brain-related phenotypes using blood *cis*-eQTL/mQTL data with large sample sizes.

#A full list of consortia members appears at the end of the paper.

G enome-wide association studies (GWAS) have discovered thousands of genetic variants associated with complex traits and diseases[1–3]. Most trait-associated variants reside in non-coding regions of the genome[4,5], suggesting that genetic variants may affect the trait through regulation of gene expression[6,7]. With the advances in microarray and sequencing technologies, genome-wide genotype and gene expression data available from relatively large samples have been generated to identify genetic variants affecting transcription abundance[8–10], i.e., expression quantitative trait loci (eQTLs). Current eQTL studies are biased toward the most accessible tissues (e.g., blood), which are often not the most relevant tissues to the traits and diseases of interest. The Genotype-Tissue Expression (GTEx) project[11–13] provides a comprehensive resource of data to investigate the genetic variation of gene expression across a broad range of tissues and cell types. Recent studies have utilized the GTEx data to demonstrate that genetic correlation of gene expression between tissues in local regions (i.e., ±1Mb of the transcription start site) is much higher than that in distal regions[14], consistent with the conclusions from the latest GTEx release[13], and that there is no evidence for the tissue-relevant eQTLs being enriched for associations with complex traits[15].

For studies that integrate GWAS results with eQTL or DNA methylation QTL (mQTL) data to identify putative functional genes and regulatory elements for brain-related phenotypes and diseases[16,17], the statistical power is limited by the small sample sizes of the brain eQTL or mQTL data (often in the order of 100s). On the other hand, there are blood eQTL and mQTL data available from thousands of individuals[8,9] and the sample sizes of some of the ongoing projects have reached 10,000s (e.g., the GoDMC and eQTLGen consortia). The questions are to what extent the cis-genetic effects on gene expression and DNA methylation (DNAm) in blood differ from those in brain and whether we can gain power to detect associations of genes (or DNAm sites) with brain-related traits by using the cis-eQTL (or cis-mQTL) effects estimated from a large blood sample as proxies for those in brain. Liu et al.[14] extended the stratified linkage disequilibrium (LD) score regression method to estimate genetic correlation ($r_g$) of gene expression between tissues at all SNPs in local or distal regions and showed that the mean estimate of pairwise $r_g$ at all local SNPs (i.e. cis-genetic correlation) was ~0.75 in 11 GTEx tissues but they did not estimate $r_g$ between brain and blood. In this study, we use a summary-data-based method to estimate the correlation of effect sizes of the top-associated cis-eQTLs (or cis-mQTLs) between blood and brain for genes expressed (or CpG sites methylated) in both tissues, accounting for errors in their estimated effects. We demonstrate by simulation and analysis of real data the gain of power by using cis-eQTL or cis-mQTL effects estimated in blood as proxies of those in brain to identify putative functional genes for brain-related complex traits and diseases.

## Results

**Correlation of cis-eQTL effects between brain and blood**. To quantify the similarity of genetic effects at the top-associated cis-eQTLs (or cis-mQTLs) between two tissues, we used a summary-data-based approach to estimate the correlation of cis-effects between two tissues ($r_b$) correcting for errors in the estimated cis-eQTL (or cis-mQTL) effects and sample overlap (Supplementary Fig. 1 and Methods). We showed by simulation (Supplementary Note 1) that $r_b$ is a good estimator of correlation of the true values of cis-genetic effects (Supplementary Fig. 2). Note that the $r_b$ method is distinct from the Spearman or Pearson correlation approach[13] because the latter does not account for errors in the estimated eQTL effects and thereby leads to an underestimation

of the correlation of true eQTL effects. We applied our method to estimate $\hat{r}_b$ at the top cis-eQTLs between different brain regions and between brain and blood in one data set, and between brain and blood in two data sets using summary-level data from GTEx v6 (whole blood and 10 brain regions)[11], the CommonMind Consortium (CMC; dorsolateral prefrontal cortex)[18], the Religious Orders Study and Memory and Aging Project (ROSMAP)[19], and the Brain eQTL Almanac project (Braineac; 10 brain regions)[20] (Methods and Supplementary Table 1). All eQTL effects were re-scaled based on the expression level per gene in standard deviation (SD) units. For the GTEx, CMC and ROSMAP data, which are based on RNA sequencing (RNA-Seq), we matched the data sets by Ensembl Gene IDs. For the Braineac data that are based on gene expression microarray, we matched the data sets by gene symbols and removed genes tagged by multiple gene expression probes to ensure a one-to-one match for genes between data sets. The main aim of this study is to quantify the extent to which cis-eQTL data in blood can be used in the SMR analysis[21] (see below) to identify genes associated with brain-related phenotypes and disorders. If we had selected the top-associated cis-eQTLs in blood and compared their effects with those in brain, we would likely suffer a form of winner's curse. To avoid the potential ascertainment bias, we selected the top cis-eQTLs in a reference tissue, i.e., GTEx-muscle ($n = 361$) or CMC ($n = 467$; independent of GTEx), using a stringent P-value threshold that is required for the SMR analysis[21] (see below), and estimated $r_b$ between brain and blood using these SNPs (Supplementary Fig. 3). Although this strategy uses only a quarter of all genes, the estimates of $r_b$ should be valid (see below). Note that the estimates of local and distal $r_g$ at all SNPs[14] would be more informative for other gene-trait association methods such as TWAS[22] and MetaXcan[23] that use all SNPs in a prediction analysis framework. We chose SMR (URLs) because of one of its features (i.e., the HEIDI test) to filter out associations due to linkage[21].

We selected the top-associated cis-eQTLs at $P_{eQTL} < 5 \times 10^{-8}$ for 4257 genes in GTEx-muscle and matched the selected genes with those in the other data sets (the number of matched genes ranged from 1113 to 3841) (Supplementary Table 2, i.e., up to 90%, with the lower numbers matched representing data sets with gene expression data for fewer genes). Note that all the matched genes were expressed in both tissues (i.e., genes which have at least 10 samples with reads per kilobase per million mapped reads (RPKM) > 0.1 and raw read counts >6)[13]. It should also be noted that our analysis below shows that the test-statistic for the difference in gene expression between tissues was almost independent of the test-statistic for the difference in SNP effect on gene expression between tissues, therefore selecting genes by cis-eQTL P-values would not bias mean gene expression in any specific tissue. We used the Jackknife approach that removes one gene at a time to estimate the sampling variance of $\hat{r}_b$ (Methods) assuming the estimated top cis-eQTL effects for different genes are independent. This assumption was approximately met given the small LD correlations among the 4257 cis-eQTLs and the subtle difference between the mean Jackknife sampling variance and the observed sampling variance in simulation (Supplementary Fig. 4).

Results showed that the effects of the top-associated cis-eQTLs were highly correlated between all the brain regions in GTEx after correcting for estimation errors, with a mean $\hat{r}_b$ of 0.94 (s.e. = 0.004; Fig. 1). These estimates are higher than the Spearman correlation estimates reported in a previous study[13] because the Spearman correlation does not account for errors in the estimated SNP effects and therefore underestimates the correlation of true effects especially when the sample size is small. The two cerebellum measures ("brain cerebellar hemisphere" and "brain

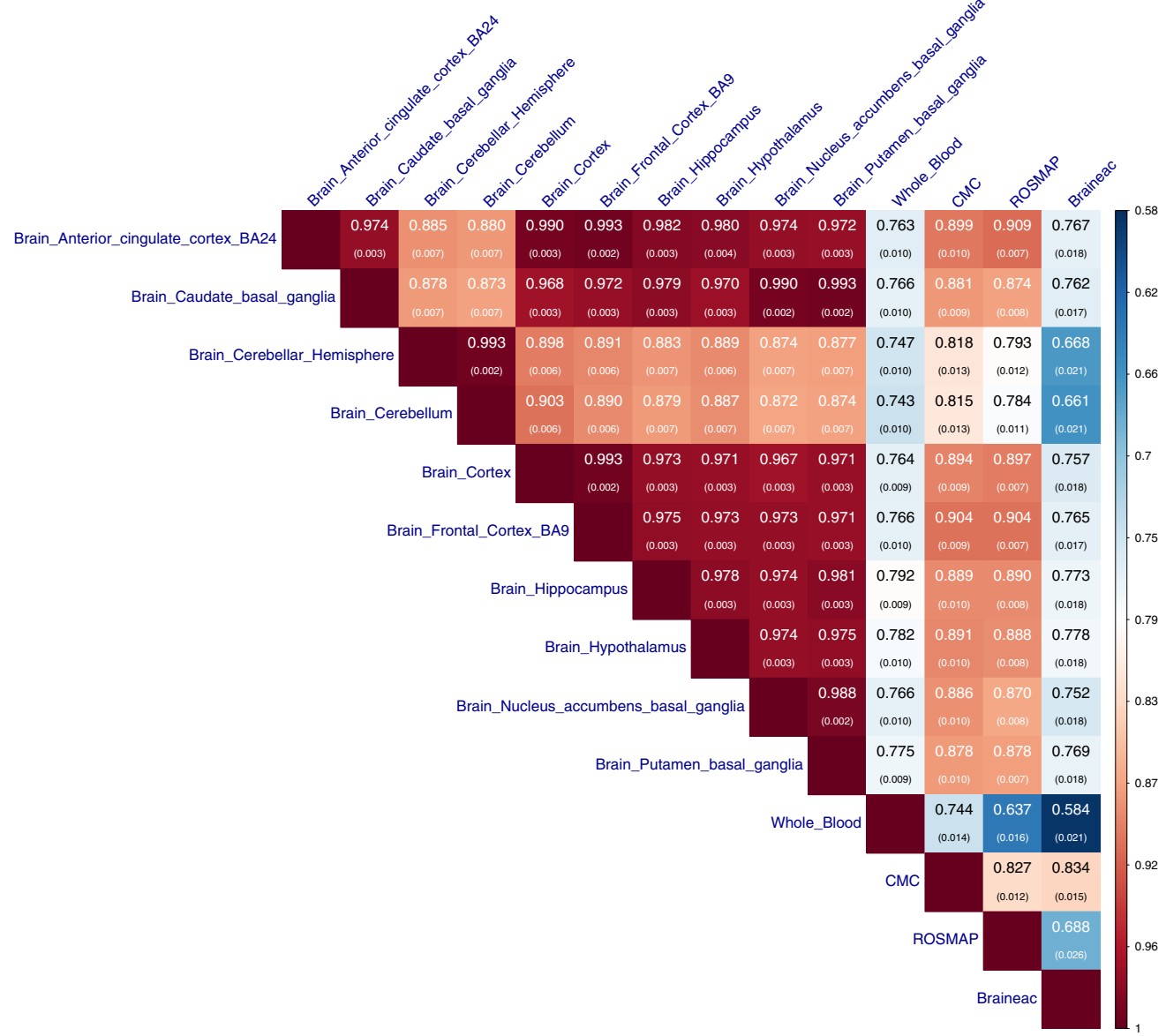

**Fig. 1** Estimated correlation of genetic effects of *cis*-eQTLs between tissues. We estimated $r_b$ between brain regions, between brain and blood tissues, and between data sets. The top-associated *cis*-eQTLs (one for each gene) were selected from GTEx-muscle at $P_{eQTL} < 5 \times 10^{-8}$. Shown in each cell is the estimate of $r_b$ with its standard error given in the parentheses (Methods). In the Braineac data, the eQTLs effect sizes were estimated from gene expression levels averaged across 10 brain regions

cerebellum") appeared to be outliers. The correlation between "brain cerebellar hemisphere" and "brain cerebellum" was almost perfect ($\hat{r}_b = 0.99$ and s.e. = 0.002), but the correlations between the two cerebellum regions and the other regions (mean $\hat{r}_b = 0.89$ and s.e. = 0.006) were significantly smaller than the pairwise correlations between the other regions (mean $\hat{r}_b = 0.98$ and s.e. = 0.003). We performed the same analysis in the Braineac data and observed similar results as above (Supplementary Fig. 5). The estimates of $r_b$ between brain and blood in GTEx varied from 0.74 to 0.79 across different brain regions with a mean estimate of 0.77 (s.e. = 0.010), similar to the mean estimate of local $r_g$ between GTEx-blood and 10 other non-brain GTEx tissues reported in a previous study[14]. The estimate of $r_b$ between CMC (brain) and GTEx-blood was 0.74 (s.e. = 0.014), suggesting that the between-sample genetic heterogeneity is small, in line with the strong correlations between CMC and GTEx brain regions (mean $\hat{r}_b =$

0.87 and s.e. = 0.010). The estimates of $r_b$ from ROSMAP were remarkably similar to those from CMC, providing an important replication of the result. The correlations related to Braineac were notably lower than those related to CMC (Fig. 1), which is likely due to the difference in transcriptomics technology between the two studies (microarray vs. RNA-Seq). It is of note that the results were robust to scale transformation of the eQTL effects (Supplementary Fig. 6), the exclusion of *cis*-eQTLs in or near the promoter regions (Supplementary Fig. 7), the exclusion of housekeeping genes[24,25] (Supplementary Fig. 8), the inclusion of secondary *cis*-eQTLs identified from a conditional analysis[26] (Supplementary Fig. 9), or the adjustment of gene expression data for confounding (e.g., batch effects) predicted from the data (Supplementary Fig. 10). In addition, we selected the top-associated *cis*-eQTLs at $P_{eQTL} < 5 \times 10^{-8}$ from the CMC data, and found that the estimates of $r_b$ among the brain regions and

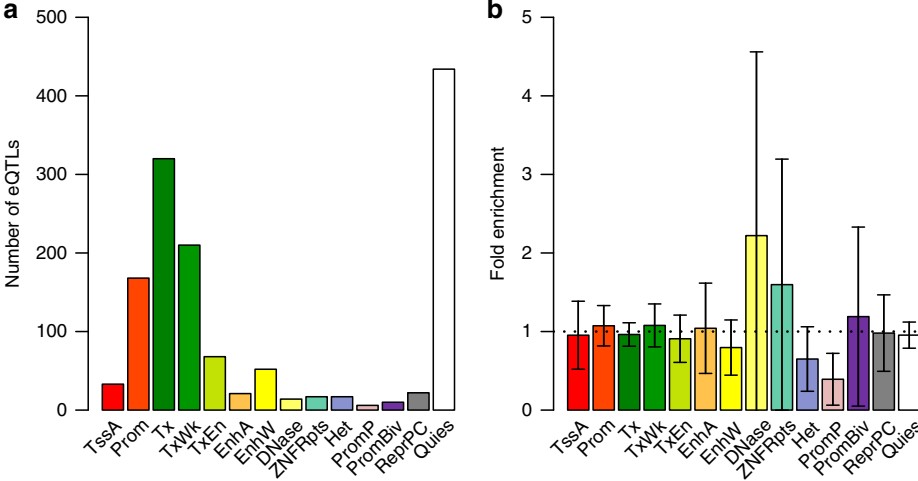

**Fig. 2** Enrichment of tissue-specific *cis*-eQTLs in functional annotations. **a** The distribution of *cis*-eQTLs across 14 functional categories derived from RMEC (Methods). **b** Estimated enrichment of $T_D$ (testing for the difference in *cis*-eQTL effect between CMC-brain and GTEx-blood) in each functional category (Methods). Error bars represent 95% confidence intervals around the estimates. The black dash line represents fold enrichment of 1. Different colors in **a** and **b** correspond to 14 functional categories: TssA: active transcription start site, Prom: upstream/downstream TSS promoter, Tx: actively transcribed state, TxWk: weak transcription, TxEn: transcribed and regulatory Prom/Enh, EnhA: active enhancer, EnhW: weak enhancer, DNase: primary DNase, ZNF/Rpts: state associated with zinc finger protein genes, Het: constitutive heterochromatin, PromP: poised promoter, PromBiv: bivalent regulatory states, ReprPC: repressed Polycomb states, and Quies: a quiescent state

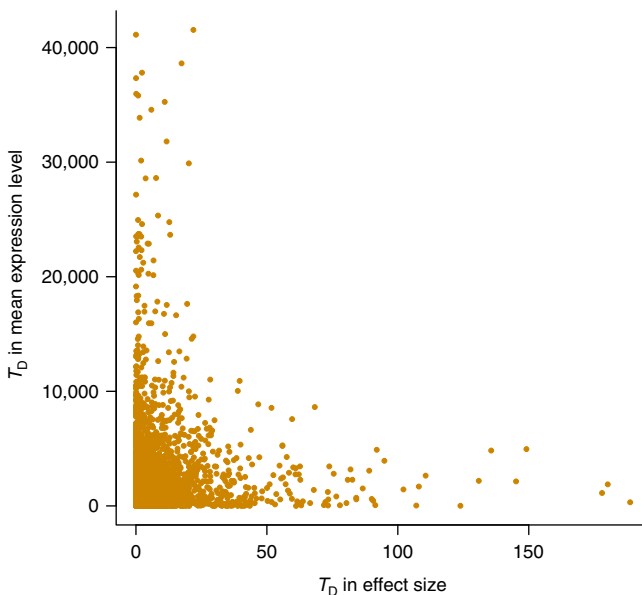

**Fig. 3** Correlation of difference in *cis*-eQTL effect and difference in expression level. Each dot represents one of the 3569 genes between GTEx-cerebellum and GTEx-blood. The 3569 genes were ascertained with at least one *cis*-eQTL with $P_{eQTL} < 5 \times 10^{-8}$ in GTEx-muscle and expressed in GTEx-cerebellum and GTEx-blood (i.e. genes which have at least 10 samples with RPKM >0.1 and raw read counts >6). In this analysis, we used *cis*-eQTL effects in SD units and gene expression levels in $\log_2$(RPKM) units to avoid confounding of the correlation by the mean–variance relationship in gene expression

between brain and blood in GTEx remained largely unchanged (Supplementary Fig. 11), suggesting that our results are also robust to the ascertainment of the *cis*-eQTLs.

**cis-eQTLs with tissue-specific effects**. The strong correlation of *cis*-eQTL effects between brain and blood (Fig. 1) does not preclude eQTLs with detectable difference in effect size between tissues. Of the 1388 *cis*-eQTLs with $P_{eQTL} < 5 \times 10^{-8}$ in GTEx-

muscle and available in CMC and GTEx-blood (Supplementary Table 2), 308 (22%) showed significant difference in effect between CMC and GTEx-blood after Bonferroni correction for multiple testing ($P_{difference} < 0.05/1388$) (Methods). Note that the substantial proportion of eQTLs with significant between-tissue differences in effect does not contradict the large estimate of $r_b$ above (Fig. 1) because the power to detect a difference in effect depends on sample size[13] (Supplementary Fig. 12). Previous studies have indicated that chromatin state at promoters is largely invariant across diverse cell types whereas enhancers are marked with highly cell-type-specific histone-modification patterns[27], that functional variants (predicted by chromatin activity data) in enhancers are less likely to be shared across many tissues compared with those in promoters[28], and that cell-type-specific eQTLs are more dispersedly distributed around the transcription start site than eQTLs affected expression in multiple cell types[29,30]. These results seem to indicate that tissue-specific eQTLs are enriched in distal regulatory elements (i.e., enhancers). To address this hypothesis, we computed the statistics to test for the between-tissue difference in eQTL effect (denoted by $T_D$) and tested the inflation (or deflation) of mean $T_D$ of *cis*-eQTLs in the functional categories annotated by the Roadmap Epigenomics Mapping Consortium (REMC)[31] (Methods). The result showed that although *cis*-eQTLs are enriched in genomic regions of active chromatin state (e.g., promoters and enhancers) and deflated in inactive regions, the mean $T_D$ of *cis*-eQTLs between CMC and GTEx-blood was almost evenly distributed across all the functional categories with no evidence of inflation in the enhancer regions (Fig. 2). The result remained largely unchanged if we repeated the enrichment analysis based on $T_D$ between GTEx-cerebellum and GTEx-blood (Supplementary Fig. 13). Note that these results do not contradict the observation from a recent study that eQTLs detected in specific tissues in GTEx tend to be most enriched among the variants predicted to be functional in relevant REMC tissues[28]. There were some examples where the *cis*-eQTLs with tissue-specific effects in brain and blood were located in enhancers (Supplementary Fig. 14). These examples, however, were rare because only 14 of the 308 eQTLs with $P_{difference} < 0.05/1388$ were located in enhancers and only 4 of the 14 enhancers appeared to be tissue specific.

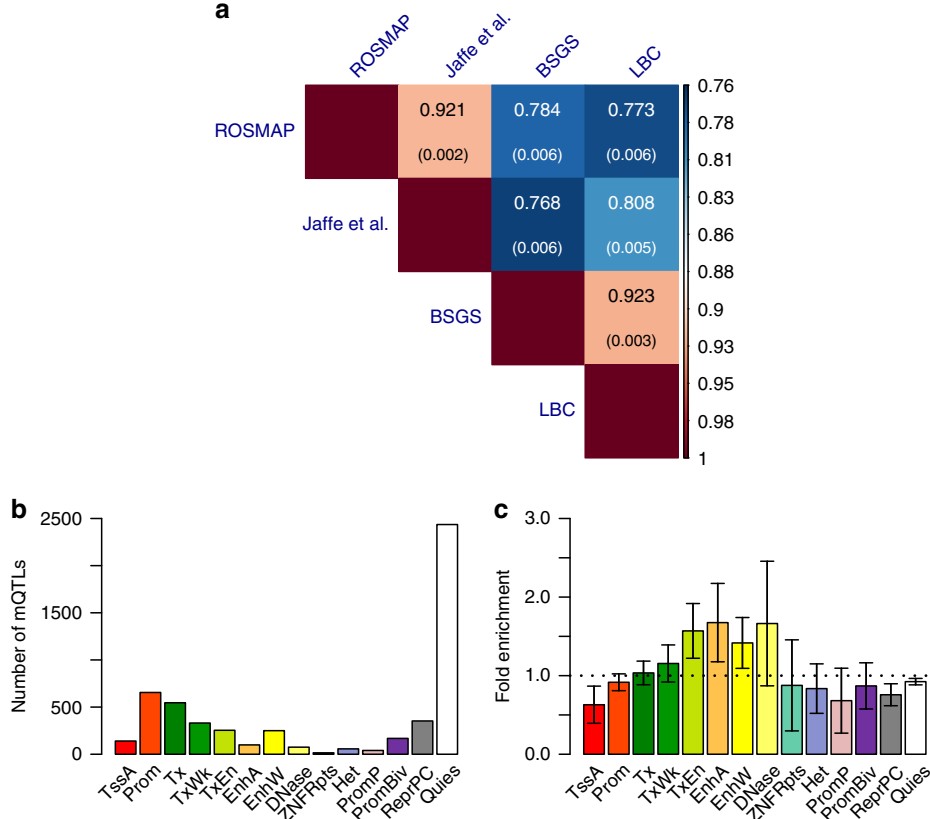

**Fig. 4** Similarity and difference in *cis*-mQTL effects between brain and blood. **a** Estimated $r_b$ for *cis*-mQTLs between brain and blood from four independent data sets. The *cis*-mQTLs (one for each DNAm probe) were selected at $P_{\mathrm{mQTL}} < 1 \times 10^{-10}$ using data from the Hannon et al. study. Shown in each cell is the estimate of $r_b$ with its standard error given in the parentheses (Methods). **b** The distribution of *cis*-mQTLs across 14 functional categories derived from RMEC (Methods). **c** Estimated enrichment of $T_D$ (testing for the difference in *cis*-mQTL effect between Jaffe-brain and LBC-blood) in each functional category (Methods). Error bars represent 95% confidence intervals around the estimates. The black dash line represents the fold enrichment of 1

In addition, there are a large number of genes showing differences in expression level between tissues[11]. It is not clear whether these differences are partly driven by the differences in eQTL effect. We sought to address this question by examining the correlation between test-statistic for difference in *cis*-eQTL effect (in SD units) and test-statistic for difference in mean expression level of the corresponding gene (in $\log_2$(RPKM) units) between GTEx-cerebellum and GTEx-blood for the 3569 genes each with a *cis*-eQTL at $P_{\mathrm{eQTL}} < 5 \times 10^{-8}$ in GTEx-muscle (Supplementary Table 2). Note that the *cis*-eQTL effects were re-scaled based on the expression level per gene in SD units so that the correlation was not confounded by the mean–variance relationship in gene expression. That is, if the difference in eQTL effect and that in expression level were both computed in RPKM units, genes with larger differences in mean between tissues are more likely to have differences in variance because of the mean–variance relationship, giving rise to differences in eQTL effect even if the eQTL effects are not different in SD units. We found that the correlation was marginal ($r = 0.003$) (Fig. 3). This is analogous to the observation that there is a large difference in mean height between men and women but the effects of all autosomal SNPs on height in men are almost identical to those in women[32,33]. However, these results also suggest that an eQTL with identical effect on gene expression in SD units in different tissues could show different effects in RPKM units if the variance of gene expression varies across tissues, which might explain the results from recent studies that genetic variants in or near genes differentially expressed in a particular tissue are enriched for associations with a complex trait[34,35].

**Correlation of *cis*-mQTL effects between brain and blood.** Having shown that *cis*-eQTL effects are highly correlated between brain and blood, we then turned to estimate the correlation of genetic effects on DNAm between the two tissues by applying the $r_b$ method to mQTL data. We analyzed summary-level mQTL data from five studies based on the Illumina HumanMethylation450K array: fetal brain from Hannon et al. ($n = 166$)[36], brain cortical region from ROSMAP ($n = 468$)[19], frontal cortex region from Jaffe et al. ($n = 526$)[37], and peripheral blood from McRae et al. (LBC: $n = 1366$ and BSGS: $n = 614$)[38] (Supplementary Table 3). All the mQTL effects are in SD units. We matched the SNPs in common across data sets, selected the top-associated *cis*-mQTL at $P_{\mathrm{mQTL}} < 1 \times 10^{-10}$ for 26,840 DNAm probes in the data from Hannon et al. (because only SNPs with $P_{\mathrm{mQTL}} < 1 \times 10^{-10}$ are available in this data set) and matched the selected probes with those in the other data sets (the number of matched probes ranged from 4892 to 6561) (Supplementary Table 4). The correlation of *cis*-mQTL effects between two brain samples (ROSMAP and Jaffe et al.) was very strong ($\hat{r}_b = 0.92$ and s.e. = 0.002), similar to that between two blood samples ($\hat{r}_b = 0.92$ between BSGS and LBC with s.e. = 0.003) (Fig. 4a). It is of note that both estimates of $r_b$ were smaller than unity, reflecting some degree of heterogeneity between studies. The mean brain–blood $r_b$ estimate from two samples was 0.78 (s.e. = 0.006) (Fig. 4a), higher than that for *cis*-eQTLs (mean $\hat{r}_b = 0.70$ and s.e. = 0.015) shown above (Fig. 1). The result remained largely unchanged if the *cis*-mQTLs were selected at $P_{\mathrm{mQTL}} < 5 \times 10^{-8}$ in the LBC data (Supplementary Fig. 15), again showing the robustness of our results to the choice of reference tissue. In addition, of the 5416

*cis*-mQTLs, 1847 (34%) showed significantly different effects between brain (Jaffe et al.) and blood (LBC) after correcting for multiple testing ($P_{difference} < 0.05/5416$). We then tested whether *cis*-mQTLs in any of the REMC functional categories tend to have higher $T_D$ between brain and blood (see above). There were small but significant enrichments of $T_D$ in enhancer regions (e.g., transcribed enhancer, active enhancer and weak enhancer) (Fig. 4c), and one of them survived multiple-testing correction (Supplementary Table 5).

**Meta-analysis of brain eQTL data from correlated samples.** We know from the $r_b$ analysis above that *cis*-eQTLs are almost perfectly correlated in different brain regions. We then sought to combine data from the brain regions to increase the power of detecting eQTLs for follow-up analysis (e.g., identification of putative functional genes for brain-related traits and diseases). However, if there is sample overlap between two tissues and the phenotypic correlation is nonzero, the estimation errors of the SNP effects from the two tissues will be correlated. We implemented in the SMR software package (URLs) a summary-data-based method, which only requires summary-level data in the *cis*-regions to account for sample overlaps, to meta-analyze *cis*-eQTL data in correlated samples (MeCS) (Methods). MeCS is very similar to existing meta-analysis approaches such as MTAG[39] or the Han et al. method[40] that account for sample overlaps. However, there is a small but important distinction. That is, MeCS uses "null" SNPs (e.g., $P_{eQTL} > 0.01$) to quantify sampling correlation of the estimated SNP effects between two data sets ($\theta$), similar to the strategy used in the latest version of METAL

(method unpublished, URLs), whereas MTAG[39] uses $\hat{\theta}$ estimated by the intercept of bivariate LD score regression[41] that relies on the assumption of an infinitesimal model which is invalid in *cis*-eQTL regions[42]. Han et al.[40] suggest the use of the number of overlapping individuals[43] or *z*-statistics to compute $\hat{\theta}$ for summary-data-based analysis. However, a meta-analysis of *cis*-eQTL effects from two tissues requires the correlation of expression level between the tissues (because $\theta = r_p\rho$ with $r_p$ being the correlation of expression level and $\rho$ being the proportion of sample overlap[44]) which is not available in summary data, and $\hat{\theta}$ estimated by the correlation of *z*-statistics in the *cis*-region could be biased by the strong local genetic correlation[14]. We showed by simulations that $\hat{\theta}$ could be estimated with high accuracy from summary data of the "null SNPs" in *cis*-region using a simple correlation approach (Supplementary Note 1, Supplementary Figs. 16 and 17), that the MeCS test-statistics were well calibrated under the null hypothesis (Supplementary Fig. 16), and that the MeCS estimates of meta-analysis effect sizes were well estimated under the alternative hypothesis (Supplementary Fig. 17). We compared MeCS to a univariate analysis of the mean expression phenotype across tissues and found that the estimates of effect size and SE from the two approaches were highly consistent (Supplementary Fig. 18). Note that in comparison with the separate analysis in individual tissues, the gain of power for MeCS increased with the decrease of correlation in expression phenotype between tissues, more so for meta-analysis using individual-level data (Supplementary Fig. 19).

We applied MeCS to data from 10 brain regions in GTEx (we referred to the meta-analyzed data as GTEx-brain hereafter).

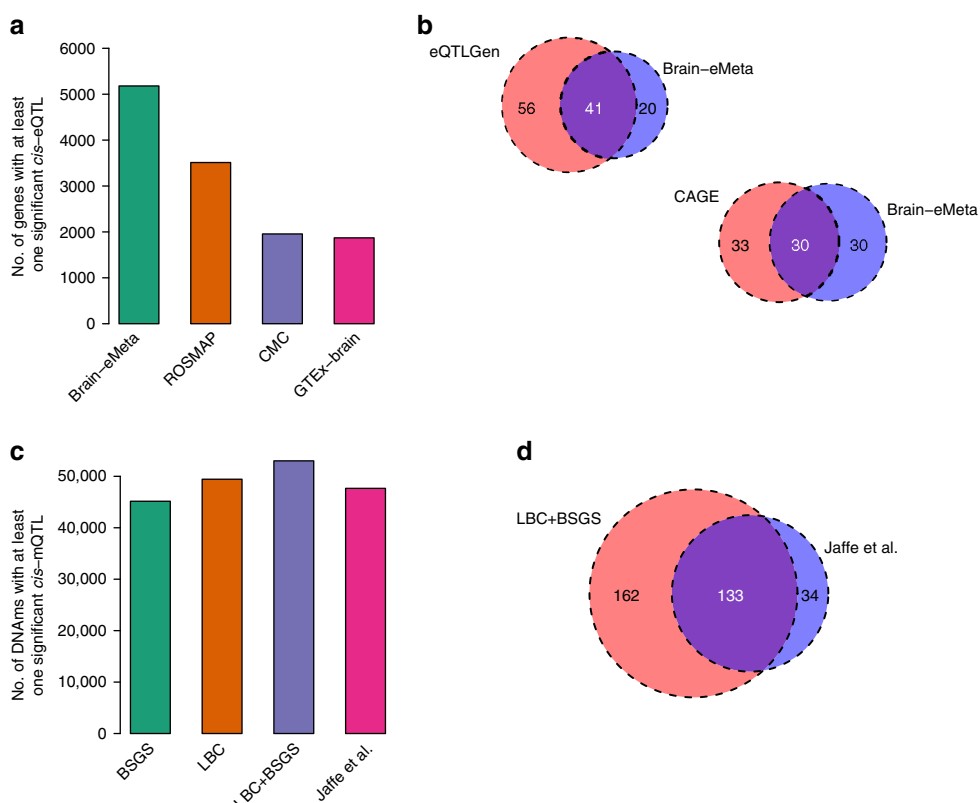

**Fig. 5** Identification of genes and DNAm sites associated with four brain-related traits. Genes (DNAm sites) associated with the brain-related traits were identified by a SMR analysis of GWAS data with eQTL (mQTL) data from brain and blood samples. The four brain-related traits are smoking, IQ, SCZ, and EduYears. **a**, **c** show the number of genes (DNAm sites) with at least one significant SNP at $P < 5 \times 10^{-8}$ in different data sets. **b**, **d** show the number of genes (DNAm sites) associated with traits identified in different data sets. Sample sizes of the brain studies: GTEx-brain ($n = \sim233$), CMC ($n = 467$), ROSMAP ($n = 494$), Brain-eMeta ($n_{eff} = \sim1194$), and Jaffe et al. ($n = 526$). Sample sizes of the blood studies: CAGE ($n = 2765$), eQTLGen ($n = 14,115$), LBC + BSGS ($n = 1980$)

 

There were strong sample overlaps among the ten brain regions (mean overlap = 70.4%) and the mean correlation in expression level between pairwise brain regions across all the expressed genes was moderate (mean $r_p = 0.33$). The gain of power by the meta-analysis was demonstrated by the observation that the mean $\chi^2$ statistic for cis-eQTLs (selected from GTEx-blood at $P_{eQTL} < 5 \times 10^{-8}$) in GTEx-brain was larger than that in any individual brain region (Supplementary Fig. 20c). The association test-statistic for a SNP can be written as $\chi^2 = 1 + n_{eff} \frac{q^2}{1-q^2}$, where $n_{eff}$ is the effective sample size and $q^2$ is the variance explained by a SNP[45]. We therefore can approximately estimate $n_{eff}$ of GTEx-brain assuming constant mean $q^2$ across brain regions (Supplementary Note 2). Note that this assumption is justified by the highly consistent estimates of variance of cis-eQTL effects across genes in different brain regions (Supplementary Fig. 21) along with a mean $r_b$ estimate of 0.94 between pairwise brain regions for cis-eQTL effects in SD units. The estimate of $n_{eff}$ of GTEx-brain was 233, approximately 2.6 times larger than the actual sample size of brain tissue in GTEx (mean $n = \sim 89$ across 10 brain regions) (Supplementary Fig. 20d). To further increase the power of detecting brain eQTLs, we meta-analyzed GTEx-brain, CMC, and ROSMAP (referred to as Brain-eMeta hereafter). The gain of power is demonstrated by the increased number of genes with at least one cis-eQTL with $P_{eQTL} < 5 \times 10^{-8}$ in Brain-eMeta as compared with that in GTEx-brain, CMC, or ROSMAP (Fig. 5a).

**Identifying DNAm and genes for brain-related phenotypes.** With the Brain-eMeta eQTL data ($n_{eff} = 1194$) obtained from the meta-analysis above, we applied the SMR approach[21,46] to test for associations of gene expression levels with four brain-related phenotypes, i.e., ever-smoked (smoking), fluid intelligence score (IQ), years of education (EduYears), and schizophrenia (SCZ). GWAS data were from published meta-analyses for EduYears and SCZ[47,48], and from analyses of the full release of the UK Biobank data for smoking and IQ (Methods and Supplementary Table 6). LD data required for the HEIDI test[21] were estimated from genotyped/imputed data of the Health and Retirement Study (HRS)[49]. LD $r^2$ from HRS were strongly correlated with those from CMC (Supplementary Fig. 22), consistent with the observation from previous studies[26]. For power comparison, we included in the SMR analysis an additional set of blood eQTL data from a sample of 14,115 individuals from the eQTLGen Consortium. Only the genes with at least one cis-eQTL at $P_{eQTL} < 5 \times 10^{-8}$ (one of the basic assumptions of SMR) in both Brain-eMeta and eQTLGen were included. We further excluded genes in the major histocompatibility complex (MHC) region because of the complexity of this region, leaving 3943 genes for analysis. We identified 61 genes associated with the traits using the brain eQTL data, 41 of which (67.2%) were in common with a larger set of genes (97) identified using the eQTLGen blood eQTL data (Fig. 5b). Despite the heterogeneity between the two eQTL data sets (Brain-eMeta was based on RNA-Seq and eQTLGen was based on microarray), the strong overlap between the two sets of results is consistent with the strong correlation of eQTL effects between brain and blood estimated above. For SCZ, 19 out of the 24 genes identified using brain eQTL data were replicated using blood eQTL data with an additional 27 genes identified only in the blood data because of its larger sample size (Supplementary Fig. 23). We repeated the SMR analysis using blood eQTL data from the Consortium for the Architecture of Gene Expression (CAGE; $n = 2765$)[9] and observed similar results (Fig. 5b) although the power of CAGE was lower than that of eQTLGen (63 genes identified using CAGE versus 97 genes identified using eQTLGen).

We also performed the SMR analysis to detect associations between DNAm sites and the brain-related phenotypes[16] using brain mQTL data from Jaffe et al. ($n = 526$) and blood cis-mQTL data from a meta-analysis of LBC and BSGS ($n = 1980$) (Methods). We only included in the analysis DNAm probes with at least one cis-mQTL with $P_{mQTL} < 5 \times 10^{-8}$ in both the brain and blood data sets. We identified 167 DNAm sites associated with the traits ($P_{SMR} < 1.8 \times 10^{-6}$) using the brain mQTL data, 133 of which (79.6%) were in common with the set of 295 DNAm sites identified using the blood mQTL data (Fig. 5d and Supplementary Fig. 24). The brain to blood "replication" rate slightly decreased when we rejected the associations with $P_{HEIDI} < 0.05$ (Supplementary Fig. 25), likely because of the HEIDI test being over-conservative especially as sample size increases[21]. These results further demonstrate the feasibility and gain of power of using the cis-genetic effects on gene expression or DNAm estimated in blood to identify putative target genes and regulatory DNA elements for brain-related phenotypes.

**Discussion**

We estimated the correlation ($\hat{r}_b$) of genetic effects at the top-associated cis-eQTLs/mQTLs between brain and blood. Because the $r_b$ method accounts for estimation errors, $\hat{r}_b$ can be interpreted as an estimate of correlation of true cis-eQTL effects between brain and blood, as demonstrated by simulations (Supplementary Fig. 2). We applied the method to summary-level eQTL data from GTEx and found that genetic effects on gene expression at the top-associated cis-eQTLs were almost perfectly correlated between different brain regions (mean $\hat{r}_b = 0.94$ for cis-eQTLs), especially between the non-cerebellar regions (mean $\hat{r}_b = 0.98$ and s.e. = 0.003), in contrast to the modest phenotypic correlation in gene expression levels (mean $r_p = 0.33$). It is therefore sensible to run a meta-analysis of the cis-eQTL effects across brain regions to gain power of detecting eQTLs for the whole brain (Supplementary Fig. 19). This can be done even if the brain regions are from different samples. We also found that the cis-eQTL effects were highly correlated between brain and blood in GTEx (mean $\hat{r}_b = 0.77$ for cis-eQTLs), and the estimate only slightly decreased using data from different samples (mean $\hat{r}_b = 0.70$). These estimates were significantly different from 1, suggesting there are real genetic differences between tissues. The genetic differences are partly due to cell-type-specific genetic effects regardless whether cell composition covariates have been included in the eQTL analysis or not. This is because adjusting for cell composition only removes the mean differences in gene expression level among cell types rather than cell-type-specific genetic effects. On the other hand, however, the strong between-tissue correlation in cis-eQTL effects does not contradict the result that many genes showed differential expression between brain and blood because the difference in cis-eQTL effect is almost independent of the mean difference in gene expression level (Fig. 3). Our results reinforce that very large sample sizes are needed to generate eQTL data sets in a specific tissue (e.g., blood) to increase the power of detecting cis-eQTLs regardless of the relative expression level of the tissue.

Our results also provide some guidelines about the use of discovery–replication paradigm to compare eQTL effects between tissues (i.e., detecting eQTLs in one tissue at a stringent $P$-value threshold and replicating the effects in another tissue after correcting for multiple tests)[13,29]. Here, we often saw a low to moderate replication rate even if there is no genetic difference between the tissues. This is because the replication rate is a function of the sample size of the validation set (Supplementary Fig. 12) and the sample sizes of eQTL studies in non-blood tissues are often limited. If we apply the discovery–replication paradigm

to the GTEx data, only ~10.7% of eQTLs discovered in GTEx-muscle could be replicated in GTEx-hippocampus (although the estimates from the recent methods[50,51] based on the discovery–replication paradigm were much higher) (Supplementary Table 7), which could potentially lead to a wrong conclusion that a large proportion of cis-eQTLs are tissue specific (note that the $r_b$ estimate between the two tissues was 0.81). We therefore do not recommend the use of the discovery–replication paradigm to quantify the tissue-specific effects especially in small samples.

We applied the SMR and HEIDI methods to identify genes and DNAm sites associated with brain-related phenotypes through pleiotropy using summary data from GWAS and cis-eQTL/mQTL studies with large sample sizes ($n_{max} = 453,693$ for GWAS, $n_{max} = 14,115$ for eQTL and $n_{max} = 1980$ for mQTL). We identified a number of genes and DNAm sites that showed pleiotropic associations with the phenotypes, consistent with a plausible model that the SNP effects on the phenotypes are mediated by genetic regulation of expression levels of the target genes and/or DNAm levels at the CpG sites. We repeated the analyses using eQTL and mQTL data from brain samples with much smaller sample sizes ($n_{max} = 1194$ for eQTL and $n_{max} = 526$ for mQTL). Due to the lower power of the data sets, the number of genes or DNAm sites detected in the brain sample was much smaller than that using the blood sample (Fig. 5, Supplementary Figs. 23–25), with at least 50% of genes (DNAm sites) in common between the two sets. These results provide strong justification for the use of blood samples to discover genes related to brain phenotypes and diseases. In practice, we recommend using a blood data set with large sample size for discovery, and an additional data set from brain for replication. This paradigm is certainly applicable to other phenotypes and their related tissues.

We conclude with several caveats. First, our estimation of $r_b$ is based on genes expressed in both brain and blood (i.e., genes only expressed in one tissue were not included in the estimation). Therefore, the estimate of $r_b$ needs to be interpreted with a restriction to genes expressed in both tissues. Although only a quarter (4257) of all genes were selected in our analysis (with at least one cis-eQTL at $P_{eQTL} < 5 \times 10^{-8}$ in GTEx-muscle), up to 90% of those selected genes were expressed in both brain and blood, reflecting the high proportion of all genes expressed in both tissues. Second, we focused our analyses only on cis-eQTLs and cis-mQTLs because trans-eQTLs and trans-mQTLs data were not available in most data sets used in our study. Although most SNP-based heritability for gene expression levels are attributed to cis-eQTLs[9], trans-eQTLs may also have an important role in regulating gene expression especially for tissue-specific effects[14]. The methods developed in this study can be applied to trans-eQTL/mQTL data with minimal modification. Because the variance explained by individual trans-eQTL/mQTL is small on average[9,38], very large sample sizes (e.g., 10,000s) are required to detect trans-eQTLs to be useful for the SMR analysis[21]. Third, the $r_b$ analysis was focused on the correlation at the top-associated cis-eQTLs/mQTLs with relatively large effects (i.e., $P < 5 \times 10^{-8}$ in a reference tissue) because the SMR test only uses cis-eQTLs/mQTLs at $P < 5 \times 10^{-8}$. The estimate of $r_b$ was slightly lower for cis-eQTLs/mQTLs selected at a less stringent threshold (Supplementary Fig. 26), consistent with the observation in simulation (Supplementary Fig. 27). However, this does not change our conclusion about the use of the top-associated cis-eQTLs/mQTLs identified in a large blood sample to identify putative target genes for brain-related traits. Last but not least, the MeCS method requires the correlation of errors in the estimated SNP effects between two samples ($\theta$), which is estimated by a simple correlation approach at the null SNPs in the cis-region. This approach, however, is not applicable to eQTL or mQTL summary data that

have been ascertained by P-value. It will also be challenging to estimate $\theta$ if only a small number of cis-SNPs are available in the summary data. We therefore recommend eQTL and mQTL studies to make more cis-SNPs available without ascertainment (e.g., all the cis-SNPs in ±2Mb of a gene or DNAm probe). Despite these caveats, our findings shed light on the genetic architecture underlying the regulation of gene expression across tissues and provide important guidance for studies in the future to identify functional genes for human complex traits.

## Methods

**Summary data of *cis*-eQTL, *cis*-mQTL, and GWAS.** This study is approved by the University of Queensland Human Research Ethics Committee (approval number: 2011001173). All the analyses of eQTL/mQTL data were performed based on summary-level data. A summary description of all the data sets can be found in Supplementary Table 1, Supplementary Table 3, and Supplementary Table 6. All the samples were of European descent and the summary data available to us were derived from individual-level data that passed stringent quantify control (QC)[9,11,18–20,36–38]. The SNPs in all eQTL/mQTL data sets were from imputation of the genotyped data to the 1000 Genomes Project (1KGP) reference panels[52], and only the SNPs with MAF > 0.01 were included in analyses.

The eQTL summary-level data were from six studies, i.e., the Genotype-Tissue Expression (GTEx)[11] v6, the CommonMind Consortium (CMC)[18], Religious Orders Study and Memory and Aging Project (ROSMAP)[19], the Brain eQTL Almanac project (Braineac)[20], the Architecture of Gene Expression (CAGE)[9], and eQTLGen. In GTEx, ROSMAP, and CMC, gene expression levels were measured by RNA-Seq. Genes in GTEx and ROSMAP were annotated by GENCODE[53] v19 and v14, respectively, and genes in CMC were annotated by Ensembl. We accessed the GTEx eQTL summary statistics of ~9.3 million SNPs for ~32,000 genes in 44 tissues (including 10 brain regions) through GTEx portal (URLs). The sample sizes of GTEx tissues ranged from 70 to 361 with an average of 160. We accessed the CMC summary data from Synapse (accession: syn2759792). The CMC eQTL summary statistics (ascertained at FDR < 0.2) of ~1.1 million SNPs for 14,366 genes were derived from individual-level data in dorsolateral prefrontal cortex of 467 subjects, 209 of which were schizophrenia patients. We accessed the ROSMAP eQTL summary statistics of ~6.4 million SNPs for 12,979 genes, which were derived from individual-level data in dorsolateral prefrontal cortex of 494 subjects. We accessed the Braineac eQTL summary statistics of ~6.2 million SNPs for 25,490 genes, which were derived from data in 10 brain regions of 134 subjects free of neurodegenerative disorders[20]. The gene expression levels in Braineac were measured by Affymetrix Human Exon 1.0 ST Arrays. For blood eQTL data, we used eQTL summary data from CAGE[9] (38,624 gene expression probes and ~8 million SNPs on 2765 subjects) and eQTLGen (44,556 gene expression probes and ~10 million SNPs on 14,115 subjects). Gene expression levels in CAGE and eQTLGen were measured by Illumina gene expression arrays. We mapped the probes to genes based on the annotations provided by Illumina. The eQTL summary data available in GTEx, CAGE, and eQTLGen were from previous analyses of standardized gene expression levels with mean 0 and variance 1, whereas expression levels in the other data sets (i.e., CMC, ROSMAP, and Braineac) were not standardized. To harmonize the units across data sets, we re-scaled the effect size and standard error (SE) of each eQTL in the CMC, ROSMAP, and Braineac based on the z-statistic, allele frequency and sample size using the method described in Zhu et al.[21] so that the eQTL effects in all data sets can be interpreted in standard deviation (SD) units.

mQTL summary statistics were from five data sets: brain cortical region from ROSMAP study ($n_{ind} = 468$, $n_{probe} = 420,103$, $n_{snp} = 5$ million)[19]; fetal brain from Hannon et al. ($n_{ind} = 166$, $n_{probe} = 26,840$, $n_{snp} = 0.3$ million)[36]; frontal cortex region from Jaffe et al. ($n_{ind} = 526$, $n_{probe} = 138,917$, $n_{snp} = 1.5$ million)[37]; and peripheral blood from McRae et al.[38] (Lothian Birth Cohorts[54] (LBC): $n_{ind} = 1366$ and Brisbane Systems Genetics Study[55] (BSGS): $n_{ind} = 614$). DNAm levels in all these five studies were based on the Illumina HumanMethylation450K array. We performed a meta-analysis of LBC and BSGS, resulting in 397,621 DNAm probes and ~7.7 million SNPs. The DNAm levels of all the five studies were not standardized. We computed the effect size and SE of each mQTL from their z-statistics using the method described in Zhu et al.[21]

We included in the analysis four brain-related complex traits, i.e., ever-smoked (smoking), fluid intelligence score (IQ), years of education (EduYears), and schizophrenia (SCZ). GWAS summary statistics for EduYears ($n = 293,723$) and SCZ (36,989 cases and 113,075 controls) were from the latest meta-analyses[47,48], and summary data for smoking ($n = 453,693$) and IQ ($n = 146,819$) were from GWAS analyses of the latest release of the UK Biobank (UKB) data[56]. Quality control and imputation of the UKB data have been detailed elsewhere[56]. We used 456,426 individuals of European descent and 7,288,503 common SNPs (MAF > 0.01) imputed from the Haplotype Reference Consortium (HRC)[57] reference panel in the analysis. IQ was measured by 13 fluid intelligence questions and detailed description of the measurement can be found at the UKB website (URLs). We adjusted IQ ($n = 146,819$) by age and sex, and standardized the adjusted phenotype by rank-based inverse-normal transformation. The GWAS analyses were

performed in BOLT-LMM[58] using all 7.3 million SNPs with a subset of 0.7 million SNPs in common with HapMap3[59] used to control for population structure and polygenic effects. We used self-reported "ever-smoked" as a dichotomous phenotype for smoking (208,988 cases and 244,705 controls). We analyzed the data in BOLT-LMM based a linear model with age and sex fitted as covariates, and transformed the effect size of each SNP on the observed 0–1 scale to odds ratio (OR) using LMOR[60] (URLs).

**Correlation of *cis*-eQTL effects between tissues.** Let $\hat{b}$ be the estimated effect at the top-associated *cis*-eQTL for a gene (i.e., one SNP per gene). We can model $\hat{b}$ as

$$\hat{b} = b + e \tag{1}$$

where $b$ is the true effect and $e$ is the estimation error. We assume that $b$ and $e$ are random variables when interrogated across genes, i.e., $b \sim N(0, \mathrm{var}(b))$ and $e \sim N(0, \mathrm{var}(e))$. The covariance of the estimated *cis*-eQTL effects between tissues $i$ and $j$ across genes can be partitioned into the covariance of true *cis*-eQTL effects and the covariance of estimation errors (if there is a sample overlap), i.e.,

$$\mathrm{cov}(\hat{b}_i, \hat{b}_j) = \mathrm{cov}(b_i, b_j) + \mathrm{cov}(e_i, e_j) = \mathrm{cov}(b_i, b_j) + r_e \sqrt{\mathrm{var}(e_i)\mathrm{var}(e_j)} \tag{2}$$

where $\mathrm{var}(e_i)$ and $\mathrm{var}(e_j)$ are the variance of the estimation errors across genes in tissues $i$ and $j$, respectively, and $r_e$ is the correlation of estimation errors across genes between two tissues, i.e., $r_e = \mathrm{cor}(e_i, e_j)$. We know from Bulik-Sullivan et al.[41] and Zhu et al.[44] that $r_e \approx r_p \rho$, where $\rho = \frac{N_s}{\sqrt{N_i N_j}}$ measures the sample overlap with $N_i$ and $N_j$ being the sample sizes in tissues $i$ and $j$, respectively, and $N_s$ being the number of overlapping individuals, and $r_p$ is the correlation of gene expression levels between two tissues in the overlapping sample. If $i = j$, then $r_e = 1$ and $\mathrm{var}(b_i) = \mathrm{var}(\hat{b}_i) - \mathrm{var}(e_i)$, where $\mathrm{var}(b_i)$ is the variance of true *cis*-eQTL effects across genes in tissue $i$. We therefore can estimate the correlation of true *cis*-eQTL effect sizes across genes between tissues $i$ and $j$ as

$$\hat{r}_b = \frac{\widehat{\mathrm{cov}}(b_i, b_j)}{\sqrt{\widehat{\mathrm{var}}(b_i)\widehat{\mathrm{var}}(b_j)}} = \frac{\widehat{\mathrm{cov}}(\hat{b}_i, \hat{b}_j) - \hat{r}_e\sqrt{\widehat{\mathrm{var}}(e_i)\widehat{\mathrm{var}}(e_j)}}{\sqrt{\left[\widehat{\mathrm{var}}(\hat{b}_i) - \widehat{\mathrm{var}}(e_i)\right]\left[\widehat{\mathrm{var}}(\hat{b}_j) - \widehat{\mathrm{var}}(e_j)\right]}} \tag{3}$$

where $\widehat{\mathrm{var}}(\hat{b}_i)$ and $\widehat{\mathrm{var}}(\hat{b}_j)$ (i.e., the estimates of $\mathrm{var}(\hat{b}_i)$ and $\mathrm{var}(\hat{b}_j)$) are the observed sample variances of $\hat{b}_i$ and $\hat{b}_j$, respectively, in a set of genes, and $\widehat{\mathrm{cov}}(\hat{b}_i, \hat{b}_j)$ is the observed sample covariance between $\hat{b}_i$ and $\hat{b}_j$ in the set of genes. However, $\widehat{\mathrm{var}}(e_i)$, $\widehat{\mathrm{var}}(e_j)$ and $\hat{r}_e$ are not observable. We know that $\mathrm{SE}^2$ of $\hat{b}$ of a SNP is an estimate of the variance of $e$ over repeated experiments for a gene. We therefore can approximate $\widehat{\mathrm{var}}(e)$ by the average of $\mathrm{SE}^2$ across genes (one SNP per gene). We also know from Eq. (2) that if $b_i = b_j = 0$, $\mathrm{cov}(\hat{b}_i, \hat{b}_j) = r_e\sqrt{\mathrm{var}(e_i)\mathrm{var}(e_j)}$. Hence, $\hat{r}_e = \frac{\widehat{\mathrm{cov}}(\hat{b}_i, \hat{b}_j)}{\sqrt{\widehat{\mathrm{var}}(e_i)\widehat{\mathrm{var}}(e_j)}} = \frac{\widehat{\mathrm{cov}}(\hat{b}_i, \hat{b}_j)}{\sqrt{\widehat{\mathrm{var}}(\hat{b}_i)\widehat{\mathrm{var}}(\hat{b}_j)}} = \widehat{\mathrm{cor}}(\hat{b}_i, \hat{b}_j)$ for null SNPs, where $\widehat{\mathrm{cor}}(\hat{b}_i, \hat{b}_j)$ is the observed sample correlation between $\hat{b}_i$ and $\hat{b}_j$ in the set of genes. In practice, we computed $\hat{r}_e$ for each gene using "null" SNPs ($P_{\mathrm{eQTL}} > 0.01$) in the *cis*-region by a simple correlation approach and took the average across genes.

The sampling variance of $\hat{r}_b$ over repeated experiments can be computed via Jackknife approach leaving one gene out at a time.

$$\hat{\mathrm{V}}(\hat{r}_b)_{\mathrm{Jackknife}} = \frac{m-1}{m}\sum_t \left[\hat{r}_{b(-t)} - \hat{r}_{b(.)}\right]^2 \tag{4}$$

where $\hat{r}_{b(-t)}$ is the estimate with the $t$-th gene left out and $\hat{r}_{b(.)} = \frac{1}{m}\sum \hat{r}_{b(-t)}$. The method is derived based on eQTL data but can be applied to data from genetic studies of different types of molecular phenotypes (e.g., DNAm and histone modification).

**Enrichment of tissue-specific eQTLs in functional categories.** We used chromatin state data from 23 blood samples (T-cell, B-cell, and hematopoietic stem cells) and 10 brain samples generated by the NIH Roadmap Epigenomics Mapping Consortium (REMC)[31]. There were 25 chromatin states predicted by ChromHMM[61] based on the imputed data of 12 histone-modification marks[31]. We classified the 25 chromatin states into 14 main functional categories by combining functionally relevant annotations[62]. We tested the difference in eQTL effect for a gene between two tissues ($i$ and $j$) using the method below. Let

$$\hat{d} = \hat{b}_i - \hat{b}_j \tag{5}$$

The sampling variance of $\hat{d}$ over repeated experiments can be written as

$$\mathrm{V}(\hat{d}) = \mathrm{V}(\hat{b}_i) + \mathrm{V}(\hat{b}_j) - 2\theta\sqrt{\mathrm{V}(\hat{b}_i)\mathrm{V}(\hat{b}_j)} \tag{6}$$

where $\hat{b}_i$ and $\hat{b}_j$ are the estimated effect sizes of the top-associated *cis*-eQTL for a gene in two tissues $i$ and $j$, $\mathrm{V}(\hat{b}_i)$ and $\mathrm{V}(\hat{b}_j)$ are the sampling variances of $\hat{b}_i$ and $\hat{b}_j$, respectively, over repeated experiments, and $\theta$ is sampling correlation between $\hat{b}_i$ and $\hat{b}_j$ for the gene over repeated experiments. In practice, $\hat{\mathrm{V}}(\hat{b}_i)$ and $\hat{\mathrm{V}}(\hat{b}_j)$ can be estimated by $\mathrm{SE}^2$ of $\hat{b}_i$ and $\hat{b}_j$, and $\hat{\theta}$ can be approximated by the sample correlation between $\hat{b}_i$ and $\hat{b}_j$ across the "null" SNPs (e.g., $P_{\mathrm{eQTL}} > 0.01$) in the *cis*-region for the gene. The significance of $\hat{d}$ can therefore be assessed by a Wald test, i.e., $T_{\mathrm{D}} = \frac{\hat{d}^2}{\mathrm{var}(\hat{d})} \sim \chi_1^2$.

To test the enrichment of $T_{\mathrm{D}}$ statistics in functional annotations, we allocated the *cis*-eQTLs to the 14 functional categories described above by physical position, and calculated the mean $T_{\mathrm{D}}$ of each category. We assessed the enrichment using the inflation factor $\lambda = \frac{\overline{T}_{\mathrm{D}(i)}}{\overline{T}_{\mathrm{D}}}$, where $\overline{T}_{\mathrm{D}(i)}$ is the mean $T_{\mathrm{D}}$ of the *cis*-eQTLs in a category $i$, and $\overline{T}_{\mathrm{D}}$ is the mean $T_{\mathrm{D}}$ of all the *cis*-eQTLs. We then used the Jackknife approach (leaving one gene out at a time) described above to compute the variability of $\lambda$. Note that although we described the enrichment test method above based on *cis*-eQTLs, the method can be applied to data from genetic studies of different types of molecular phenotypes (e.g., DNAm and histone modification).

**Meta-analysis of *cis*-eQTL data from correlated samples.** We know from Eq. (1) that the estimated effect of a *cis*-eQTL for a gene can be partitioned into two components, i.e., the true effect size ($b$) and the estimation error ($e$). For multiple tissues, the joint distribution of the estimates can be written as

$$\hat{\mathbf{b}} \sim N(\mathbf{1}b, \mathbf{S}) \tag{7}$$

where $\hat{\mathbf{b}} = [\hat{b}_1, \hat{b}_2, \ldots, \hat{b}_t]$, $\mathbf{S}$ is the sampling (co)variance matrix of $\hat{\mathbf{b}}$ over repeated experiments with $S_{ij} = C(\hat{b}_i, \hat{b}_j)$. $S_{ij} = \theta_{ij}S_iS_j$ when $i \neq j$, where $\theta_{ij}$ is sampling correlation between $\hat{b}_i$ and $\hat{b}_j$ for the gene over repeated experiments. $S_i^2$ and $S_j^2$ are the sampling variance of $\hat{b}_i$ and $\hat{b}_j$, respectively, over repeated experiments. If $i = j$, then $\theta_{ij} = 1$ and $S_{ij} = S_i^2$. In practice, $\theta_{ij}$ can be approximated by the sample correlation of the estimated SNP effects between a pair of tissues across the "null" SNPs (e.g., $P_{\mathrm{eQTL}} > 0.01$) in the *cis*-region for each gene. Similar to the summary-data-based meta-analysis methods that account for correlated estimation errors[39,40,63], we can estimate combined effect as

$$\hat{b} = \left(\mathbf{1}^T\hat{\mathbf{S}}^{-1}\mathbf{1}\right)^{-1}\mathbf{1}^T\hat{\mathbf{S}}^{-1}\hat{\mathbf{b}} \tag{8}$$

$$\hat{\mathrm{V}}(\hat{b}) = \frac{1}{\mathbf{1}^T\hat{\mathbf{S}}^{-1}\mathbf{1}} \tag{9}$$

The significance of $\hat{b}$ can be assessed by a Wald test, i.e., $\frac{\hat{b}^2}{\hat{V}(\hat{b})} \sim \chi_1^2$.

**URLs.** For MeCS, see http://cnsgenomics.com/software/smr/#MeCS. For SMR, see http://cnsgenomics.com/software/smr. For LMOR, see http://cnsgenomics.com/shiny/LMOR/. For UK Biobank, see http://biobank.ctsu.ox.ac.uk/. For METAL, see https://genome.sph.umich.edu/wiki/METAL. For GTEx Portal, see http://www.gtexportal.org/. For CMC data, see https://www.synapse.org/CMC. For Braineac data, see http://www.braineac.org/.

**Data availability**. Brain-eMeta eQTL summary data are available at http://cnsgenomics.com/software/smr/#Download. The eQTLGen summary data are available through application to the eQTLGen consortium. All the other data sets used in this study are from the public domain. The software tools are available at the URLs above.

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

## Acknowledgements

This research was supported by the Australian Research Council (DP160101343, DP160101056, DP160103860, and DP160102400), the Australian National Health and Medical Research Council (1113400, 1107258, 1083656, 1078037, and 1078901), the US National Institutes of Health (GM099568, GM075091, and AG042568), and the Sylvia & Charles Viertel Charitable Foundation. This study makes use of data from dbGaP (accessions: phs000428 and phs000424), UK Biobank Resource (application number: 12514), UK10K project and CommonMind Consortium. A full list of acknowledgements to these data sets can be found in the Supplementary Note 3.

## Author contributions

J.Y. conceived and designed the study. T.Q. performed statistical analyses under the assistance and guidance from Y.W., J.Z., F.Z., A.X., L.J., Z.Z., K.K., L.Y., Z.L.Z., R.E.M., G. W.M., I.J.D., N.R.W., P.M.V., A.F.M., and J.Y. J.Y. and T.Q. wrote the manuscript with the participation of all authors.

## Additional information

**Competing interests:** The authors declare no competing interests.

Ting Qi[1], Yang Wu[1], Jian Zeng[1], Futao Zhang[1,2], Angli Xue[1], Longda Jiang[1], Zhihong Zhu[1], Kathryn Kemper[1], Loic Yengo[1], Zhili Zheng[1,3], eQTLGen Consortium, Riccardo E. Marioni[4,5], Grant W. Montgomery[1], Ian J. Deary[5], Naomi R. Wray[1,2], Peter M. Visscher[1,2], Allan F. McRae[1] & Jian Yang[1,2,3]

[1]Institute for Molecular Bioscience, The University of Queensland, Brisbane, QLD 4072, Australia. [2]Queensland Brain Institute, The University of Queensland, Brisbane, QLD 4072, Australia. [3]The Eye Hospital, School of Ophthalmology & Optometry, Wenzhou Medical University, 325027 Wenzhou, Zhejiang, China. [4]Medical Genetics Section, Centre for Genomic and Experimental Medicine, Institute of Genetics and Molecular Medicine, University of Edinburgh, Edinburgh EH4 2XU, UK. [5]Department of Psychology, Centre for Cognitive Ageing and Cognitive Epidemiology, University of Edinburgh, 7 George Square, Edinburgh EH8 9JZ, UK.

## eQTLGen Consortium

Mawussé Agbessi[6], Habibul Ahsan[7], Isabel Alves[6], Anand Andiappan[8], Philip Awadalla[6], Alexis Battle[9], Frank Beutner[10], Marc Jan Bonder[11], Dorret Boomsma[12], Mark Christiansen[13], Annique Claringbould[11], Patrick Deelen[11], Tõnu Esko[14], Marie-Julie Favé[6], Lude Franke[11], Timothy Frayling[15], Sina Gharib[16], Gregory Gibson[17], Gibran Hemani[18], Rick Jansen[12], Mika Kähönen[19,20], Anette Kalnapenkis[14], Silva Kasela[14], Johannes Kettunen[21], Yungil Kim[9], Holger Kirsten[22], Peter Kovacs[23], Knut Krohn[24], Jaanika Kronberg-Guzman[14], Viktorija Kukushkina[14], Zoltan Kutalik[25], Bernett Lee[8], Terho Lehtimäki[26], Markus Loeffler[27], Urko M. Marigorta[17], Andres Metspalu[14], Lili Milani[14], Martina Müller-Nurasyid[28], Matthias Nauck[29], Michel Nivard[12], Brenda Penninx[12], Markus Perola[21], Natalia Pervjakova[14], Brandon Pierce[7], Joseph Powell[1], Holger Prokisch[30], Bruce Psaty[31], Olli Raitakari[32,33], Susan Ring[34], Samuli Ripatti[21], Olaf Rotzschke[8], Sina Ruëger[25], Ashis Saha[9], Markus Scholz[27], Katharina Schramm[28], Ilkka Seppälä[26], Michael Stumvoll[23], Patrick Sullivan[35], Alexander Teumer[36], Joachim Thiery[37], Lin Tong[7], Anke Tönjes[38], Jenny van Dongen[12], Joyce van Meurs[39], Joost Verlouw[39], Uwe Völker[40], Urmo Võsa[11], Hanieh Yaghootkar[15] & Biao Zeng[17]

[6]Computational Biology, Ontario Institute for Cancer Research, Toronto, ON M5G 0A3, Canada. [7]Department of Public Health Sciences, University of Chicago, Chicago, IL 60637, USA. [8]Singapore Immunology Network, Agency for Science, Technology and Research, Singapore 138648, Singapore. [9]Department of Computer Science, Johns Hopkins University, Baltimore, MD 21218, USA. [10]Heart Center Leipzig, Universität Leipzig, 04289 Leipzig, Germany. [11]Department of Genetics, University Medical Centre Groningen, 9713 GZ Groningen, The Netherlands. [12]Faculty of Genes, Behavior and Health, Vrije Universiteit Amsterdam, 1081 HV Amsterdam, The Netherlands. [13]Cardiovascular Health Research Unit, University of Washington, Seattle, WA 98195, USA. [14]Estonian Genome Center, University of Tartu, 50090 Tartu, Estonia. [15]Exeter Medical School, University of Exeter, Exeter EX4 4QD, UK. [16]Department of Medicine, University of Washington, Seattle, WA 98195, USA. [17]School of Biological Sciences, Georgia Tech, Atlanta, GA 30332, USA. [18]MRC Integrative Epidemiology Unit, University of Bristol, Bristol BS8 1TH, UK. [19]Department of Clinical Physiology, Tampere University Hospital, 33521 Tampere, Finland. [20]Faculty of Medicine and Life Sciences, University of Tampere, 33100 Tampere, Finland. [21]National Institute for Health and Welfare, University of Helsinki, 00100 Helsinki, Finland. [22]Institute für Medizinische InformatiK, Statistik und Epidemiologie, LIFE–Leipzig Research Center for Civilization Diseases, Universität Leipzig, 04107 Leipzig, Germany. [23]IFB Adiposity Diseases, Department of Medicine, Universität Leipzig, 04103 Leipzig, Germany. [24]Interdisciplinary Center for Clinical Research, Faculty of Medicine, Universität Leipzig, 04103 Leipzig, Germany. [25]Lausanne University Hospital, 1011 Lausanne, Switzerland. [26]Department of Clinical Chemistry, Fimlab Laboratories and Faculty of Medicine and Life Sciences, University of Tampere, 33110 Tampere, Finland. [27]Institut für Medizinische InformatiK, Statistik und Epidemiologie, LIFE–Leipzig Research Center for Civilization Diseases, Universität Leipzig, 04103 Leipzig, Germany. [28]Institute of Genetic Epidemiology, Helmholtz Zentrum München, 81377 München, Germany. [29]Institute of Clinical Chemistry and Laboratory Medicine, University Medicine Greifswald, 17489 Greifswald, Germany. [30]Institute of Human Genetics, Helmholtz Zentrum München,

81675 München, Germany. [31]Cardiovascular Health Research Unit, Departments of Epidemiology, Medicine, and Health Services, University of Washington, Seattle, WA 98195, USA. [32]Department of Clinical Physiology and Nuclear Medicine, Turku University Hospital, 20521 Turku, Finland. [33]University of Turku, 20500 Turku, Finland. [34]School of Social and Community Medicine, University of Bristol, Bristol BS8 1TH, UK. [35]Department of Medical Epidemiology and Biostatistics, Karolinska Institute, 171 77 Solna, Sweden. [36]Institute for Community Medicine, University Medicine Greifswald, 17489 Greifswald, Germany. [37]Institute for Laboratory Medicine, LIFE–Leipzig Research Center for Civilization Diseases, Universität Leipzig, 04107 Leipzig, Germany. [38]Division of Endocrinology and Nephrology, Department of Medicine, Universität Leipzig, 04103 Leipzig, Germany. [39]Department of Internal Medicine, Erasmus Medical Centre, 3015 CE Rotterdam, The Netherlands. [40]Interfaculty Institute for Genetics and Functional Genomics, University Medicine Greifswald, 17489 Greifswald, Germany

