## [Peer Review File · Nature Communications]

Reviewer #1 (Remarks to the Author):

The revision is much better than the original manuscript, it is written more clearly, corrects errors for the most part and will speak better to readers. The same is true for the Supplement. A key observation of the manuscript is said well in the Abstract: “Here, we estimate the correlation of genetic effects at the top associated cis-expression (cis-eQTLs or cismQTLs) between brain and blood for genes expressed (or CpG sites methylated) in both tissues, while accounting for errors in their estimated effects (rb). [We] we find that the genetic effects of cis-eQTLs [] or mQTLs [] are highly correlated between independent brain and blood samples.” This is an important take home message. The field gives up a lot of statistical power inherent in massive data sets by insisting on tissue-specific analyses.

The manuscript suffers from a simple and related fact, raised by all 3 reviewers, that the analyses cannot be generalized to those genes that are not expressed across a wide array of heterogeneous tissues. The authors seem to argue otherwise, saying “there was no correlation between the test-statistics for tissue-specific gene expression and the test-statistics for tissue-specific SNP effects on gene expression ...” I do not follow the argument. Tissue specific gene expression and tissue-specific eQTLs are not analyzed here. Reviewer 3, for example, expressed concern that the genes analyzed [were likely to be] only housekeeping genes. The other two reviewers said similar things in different words. The authors can answer this concern, and easily, by analyzing their gene sets to determine how many of the selected and analyzed genes are on lists of housekeeping genes. I suspect representation is far above what we would expect from a random sample of genes. This subject needs to be addressed, in my opinion.

Reviewer #2 (Remarks to the Author):

The authors have addresses all my previous comments

Reviewer #3 (Remarks to the Author):

I appreciate the clarifications and improvements in the revised manuscript. However, I didn't find that the authors had adequately responded to my previous comments, and I still have major concerns about this manuscript.

I appreciate that the authors clarified that methods development is no longer a main contribution of this paper. As is, though, it is still quite easy to read this manuscript and conclude that estimating cis-genetic correlation is a novel goal and that 0.7 is a surprising number. The estimate of genetic correlation in Liu et al. is an estimate of cis-genetic correlation, not cis+trans genetic correlation. The correlation between the cis-genetic component for two tissues is equivalent under the model assumed there to the correlation between effect sizes in the cis region for two tissues, and so the only difference I see between the method there and the method introduced here is that here the method aims to estimate the correlation between significant effects instead of between all cis effects. Supp Fig 25 (red lines only) shows though that these are quite similar: if I understand the simulation framework, then 0.7 is the estimand of Liu et al., while 0.73 is the estimand of Qi et al. I think the reason the authors focus on significant effects is because they are interested in applying SMR, which uses significant effects; if they were going to apply a different gene prioritization method such as TWAS, they may instead be interested in the Liu et al. estimand. More clarification is needed here. When the authors first introduce their method in the main text, they should make it clear that there is another method with a very similar estimand, and that based on the results of the previous work they would expect an r_b near 0.7. The authors should also clarify the relationship between their method and the Liu et al. method, and if their method is interesting specifically because of SMR and not gene prioritization in general, then they should clarify that as well.

Similarly, the authors don't adequately address the previous work of Han et al., who introduced a method that as I understand it does exactly what MeCS does. Replacing "We proposed a summary-data-based method" with "We implemented in the SMR software tool a summary-data-based method" and then citing Han et al in the methods is not an adequate fix here. Again, the authors should honestly lay out what the state of the field was before they wrote the paper and justify what they did, in the main text. If they have introduced a new method to do just what Han et al. do, then they should compare the two methods and justify using their new method. Here, again, it would be easy to read this paper and get the impression of more novelty than there really is.

I also remain concerned about the authors' result here that differential expression does not correlate with difference in effect size. I agree with the authors that this would be an important contribution. The authors clarified that the scale of gene expression is in units of standard deviation, but this is exactly what I was worried about. In units of RPKM, there may be a very large and important correlation then! There could also be a correlation between significance of the difference in effect size, and differential expression. These must be assessed and discussed.

The authors use this result to contest the idea that differentially expressed genes are informative about phenotype-relevant tissues. I would point them to Finucane et al. 2017 bioRxiv, where there are very strong phenotype-tissue associations based on differentially expression genes. Perhaps these associations are the result of correlations between difference in effect size and differential expression on the RPKM scale? I also didn't understand how the authors reached the conclusion that "Our results reinforce the need to generate tissue-specific eQTL data sets to identify variants that generate variation between people in a specific tissue regardless of the relative expression level of the tissue" -- I thought a major point of this manuscript is that the tissue does not matter too much and we can use e.g. blood instead of brain.

The authors write that Backenroth et al did not study eQTLs, but they did; see e.g. Table 1 of Backenroth et al. Why such different results here? Please discuss.

I still find the methods confusingly written -- in particular, what depends on the SNP, what depends on the gene, what depends on the tissue, and what is fixed vs random where. More subscripts might help here, along with an indication of what is varying under "cov" and "var". As an example of what is confusing: it seems like in the first part of the methods, $\text{cov}(\hat{\beta}_i, \hat{\beta}_j)$ is observed, while in the second part of the method, it must be estimated. There should be different notation for different quantities.

A minor point: I'm not sure the authors understood my comment about a correlation of 1 not indicating equality. x and $x/2$ have a correlation of 1, even if both are in the same units.

Response to Reviewers' Comments

Reviewer #1 (Remarks to the Author):

The revision is much better than the original manuscript, it is written more clearly, corrects errors for the most part and will speak better to readers. The same is true for the Supplement. A key observation of the manuscript is said well in the Abstract: "Here, we estimate the correlation of genetic effects at the top associated cis-expression (cis-eQTLs or cismQTLs) between brain and blood for genes expressed (or CpG sites methylated) in both tissues, while accounting for errors in their estimated effects (r_b). [We] we find that the genetic effects of cis-eQTLs [] or mQTLs [] are highly correlated between independent brain and blood samples." This is an important take home message. The field gives up a lot of statistical power inherent in massive data sets by insisting on tissue-specific analyses.

Re: We thank the reviewer for the positive remarks that our revision is much better than the original manuscript and that the take-home message is important to the field.

The manuscript suffers from a simple and related fact, raised by all 3 reviewers, that the analyses cannot be generalized to those genes that are not expressed across a wide array of heterogeneous tissues. The authors seem to argue otherwise, saying "there was no correlation between the test-statistics for tissue-specific gene expression and the test-statistics for tissue-specific SNP effects on gene expression ..." I do not follow the argument. Tissue specific gene expression and tissue-specific eQTLs are not analyzed here. Reviewer 3, for example, expressed concern that the genes analyzed [were likely to be] only housekeeping genes. The other two reviewers said similar things in different words. The authors can answer this concern, and easily, by analyzing their gene sets to determine how many of the selected and analyzed genes are on lists of housekeeping genes. I suspect representation is far above what we would expect from a random sample of genes. This subject needs to be addressed, in my opinion.

Re: We thank the reviewer for this helpful suggestion. We have clarified in the manuscript (pages 3, 5, and 12) that our conclusion only applied to genes that are expressed in both brain and blood. Regarding the comment about housekeeping (HK) genes, we downloaded two lists of HK genes from Lin et al. (2017 bioRxiv) and Eisenberg et al. (2013 Trends in Genetics), respectively. The number of HK genes in our ascertained gene list is significantly higher than what we would expect from a random sample of genes (see **Figure R1** below). This is expected, as pointed out by the reviewer, because HK genes are defined as a set of genes expressed across most cell types and tissues, which are expected to be enriched in genes expressed in both brain and blood. However, the estimates of r_b remained largely unchanged if we excluded the HK genes from the analysis (**Supplementary Fig. 8**), suggesting that the estimates of r_b are robust to the inclusion/exclusion of HK genes. We have clarified this in our revised manuscript (page 6): "The results were robust to scale transformation of the eQTL effects (**Supplementary Fig. 6**), the exclusion of cis-eQTLs in or near the promoter regions (**Supplementary Fig. 7**), **the exclusion of housekeeping genes (Supplementary Fig. 8), ...**".

We agree with the reviewer that it might not be appropriate to say that "there was no correlation between the test-statistic for tissue-specific gene expression and the test-statistic for tissue-specific SNP effects on gene expression ...". We have revised the sentences as (page 5): "It should also be noted that our analysis below shows that **the test-statistic for the difference in gene expression between tissues was almost independent of the test-statistic for the difference in SNP effect on gene expression between tissues**" and as (page 7): "The lack of correlation between **test-statistic for**

difference in SNP effect on gene expression and test-statistic for difference in expression level of the corresponding gene also means that the selection of genes at $P_{\text{cis-eQTL}} < 5 \times 10^{-8}$ in muscle for the r_b analysis above was not a cause of enrichment (or depletion) of genes with tissue-specific expression”

Figure R1 The number of housekeeping (HK) genes in our ascertained gene set ($m = 4,257$) vs. that expected by chance. Shown in gray is the distribution of the numbers of HK genes in 2,000 random gene sets ($m = 4,257$ in each random set). The red dashed line represents the observed number of HK genes in our ascertained gene set. The HK gene lists were obtained from Lin et al. (2017 bioRxiv) and Eisenberg et al. (2013 Trends in Genetics).

Reviewer #2 (Remarks to the Author):

The authors have addresses all my previous comments

Reviewer #3 (Remarks to the Author):

I appreciate the clarifications and improvements in the revised manuscript. However, I didn't find that the authors had adequately responded to my previous comments, and I still have major concerns about this manuscript.

I appreciate that the authors clarified that methods development is no longer a main contribution of this paper. As is, though, it is still quite easy to read this manuscript and conclude that estimating cis-genetic correlation is a novel goal and that 0.7 is a surprising number. The estimate of genetic correlation in Liu et al. is an estimate of cis-genetic correlation, not cis+trans genetic correlation. The correlation between the cis-genetic component for two tissues is equivalent under the model assumed there to the correlation between effect sizes in the cis region for two tissues, and so the only difference I see between the method there and the method introduced here is that here the method aims to estimate the correlation between significant effects instead of between all cis effects. Supp Fig 25 (red lines only) shows though that these are quite similar: if I understand the simulation framework, then 0.7 is the estimand of Liu et al., while 0.73 is the estimand of Qi et al. I think the reason the authors focus on significant effects is because they are interested in applying SMR, which uses significant effects; if they were going to apply a different gene prioritization method such as TWAS, they may instead be interested in the Liu et al. estimand. More clarification

is needed here. When the authors first introduce their method in the main text, they should make it clear that there is another method with a very similar estimand, and that based on the results of the previous work they would expect an r_b near 0.7. The authors should also clarify the relationship between their method and the Liu et al. method, and if their method is interesting specifically because of SMR and not gene prioritization in general, then they should clarify that as well.

Re: We thank the reviewer for the additional comments.

We have made it clear in the Introduction section (page 3) of the revised manuscript that there is another method to estimate the genetic correlation of gene expression at all the SNPs in local region (i.e. the cis-genetic correlation): "...to detect associations of genes (or DNAm sites) with brain-related traits by using the cis-eQTL (or cis-mQTL) effects estimated from a large blood sample as proxies for those in brain. Liu et al. (2017 AJHG) extended the stratified linkage disequilibrium (LD) score regression method to estimate genetic correlation (r_g) of gene expression between tissues at all SNPs in local or distal regions and showed that the mean estimate of pairwise r_g at all local SNPs (i.e. cis-genetic correlation) was ~ 0.75 in 11 GTEx tissues but they did not estimate r_g between brain and blood. In this study, we use a summary-data-based method to estimate the correlation of effect sizes of the top associated cis-eQTLs (or cis-mQTLs) between blood and brain for genes expressed (or CpG sites methylated) in both tissues, accounting for errors in their estimated effects."

We have also clarified in the revised manuscript that (page 4) the main aim of this study is to quantify the extent to which cis-eQTL data in blood can be used in the SMR analysis for the identification of genes associated with brain-related phenotypes and disorders, and that (page 4) the estimates of local and distal r_g at all SNPs (Liu et al. 2017 AJHG) would be more informative for other gene-trait association methods such as TWAS (Gusev et al. 2016 Nature Genetics) and MetaXcan (Barberia et al. 2016 bioRxiv) that use all SNPs in a prediction analysis framework. We chose SMR (URLs) because of one of its features (i.e., the HEIDI test) to filter out associations due to linkage (Zhu et al. 2016 Nature Genetics).

We have also commented in the Results section (page 5) that our estimates of r_b between GTEx-blood and GTEx-brain were similar to the mean estimate of local r_g between GTEx-blood and 10 other non-brain GTEx tissues reported in a previous study (Liu et al. 2017 AJHG).

We further highlighted one of the main conclusions from Liu et al. in the Introduction section of the revised manuscript (page 3): "Recent studies have utilized the GTEx data to demonstrate that genetic correlation of gene expression between tissues in local regions (i.e. ± 1 Mb of the transcription start site) is much higher than that in distal regions (Liu et al. 2017 AJHG)".

Another clarification: in **Supplementary Fig. 27** in the revised manuscript, 0.7 is the parameter of r_b used to generate the simulated data rather than the estimate from Liu et al (2017 AJHG).

Similarly, the authors don't adequately address the previous work of Han et al., who introduced a method that as I understand it does exactly what MeCS does. Replacing "We proposed a summary-data-based method" with "We implemented in the SMR software tool a summary-data-based method" and then citing Han et al in the methods is not an adequate fix here. Again, the authors should honestly lay out what the state of the field was before they wrote the paper and justify what they did, in the main text. If they have introduced a new method to do just what Han et al. do, then they should compare the two methods and justify using their new method. Here, again, it would be easy to read this paper and get the impression of more novelty than there really is.

Re: We thank the reviewer for this comment, which allows us to highlight the feature of MeCS more explicitly.

In the Methods section of the previous manuscript, we did state that MeCS is similar to the existing summary-data-based meta-analysis methods that can account for correlated estimation errors.

In the revised manuscript, we have further clarified that (pages 8 and 9) “MeCS is very similar to existing meta-analysis approaches such as MTAG (Turley et al. 2018 Nature Genetics) or the Han et al. method (Han et al. 2016 HMG) that account for sample overlaps. However, there is a small but important distinction. That is, MeCS uses “null” SNPs (e.g. $P_{eQTL} > 0.01$) to quantify the sampling correlation of the estimated SNP effects between two data sets (θ), similar to the strategy used in the latest version of METAL (method unpublished, URLs), whereas MTAG (Turley et al. 2018 Nature Genetics) uses $\hat{\theta}$ estimated by the intercept of bivariate LD score regression (Bulik-Sullivan et al. 2015 Nature Genetics) that relies on the assumption of an infinitesimal model which is invalid in cis-eQTL/mQTL regions (Shi et al. 2016 AJHG). Han et al. (Han et al. 2016 HMG) suggest the use of the number of overlapping individuals (Lin et al. 2009 AJHG) or z-statistics to compute $\hat{\theta}$ for summary-data-based analysis. However, a meta-analysis of cis-eQTL effects from two tissues requires the correlation of expression level between the tissues (because $\theta = r_p \rho$ with r_p being the correlation of expression level and ρ being the proportion of sample overlap (Zhu et al. 2018 Nature Communications)) which is not available in summary data, and $\hat{\theta}$ estimated by the correlation of z-statistics in the cis-region could be biased by the strong local genetic correlation (Liu et al. 2017 AJHG).”

I also remain concerned about the authors' result here that differential expression does not correlate with difference in effect size. I agree with the authors that this would be an important contribution. The authors clarified that the scale of gene expression is in units of standard deviation, but this is exactly what I was worried about. In units of RPKM, there may be a very large and important correlation then! There could also be a correlation between significance of the difference in effect size, and differential expression. These must be assessed and discussed.

Re: The difference in eQTL effect between tissues was estimated using gene expression level in standard deviation (SD) units and the difference in gene expression level was computed in $\log_2(\text{RPKM})$ units because otherwise a correlation between the two is expected due to a mean-variance relationship. That is, if the difference in eQTL effect and the difference in expression level were both computed in RPKM units, genes with differences in mean between tissues will tend to have differences in variance because of the mean-variance relationship, giving rise to differences in eQTL effects even if the eQTL effects are not different in SD units. We have clarified this in the revised manuscript (pages 4 and 7).

The authors use this result to contest the idea that differentially expressed genes are informative about phenotype-relevant tissues. I would point them to Finucane et al. 2017 bioRxiv, where there are very strong phenotype-tissue associations based on differentially expression genes. Perhaps these associations are the result of correlations between difference in effect size and differential expression on the RPKM scale?

Re: Finucane et al. (2017 bioRxiv) show that genetic variants in or near genes differentially expressed in a particular tissue are enriched for associations with a complex trait. However, it is not clear whether the enrichment is due to the effects of trait-associated variants on gene expression.

Therefore, our result does not contradict that in Finucane et al. (2017 bioRxiv). We have commented on this in the revised manuscript (page 7).

We have also clarified that the aim of this correlation (between difference in effect size and differential expression) analysis is to address the question whether the between-tissue differences in gene expression are partly driven by the differences in eQTL effects (page 7) rather than the enrichment of GWAS signals in or near differentially expressed genes.

To avoid confusion, we **removed** the following statements from the manuscript.

“GWAS signals for a trait that are located in or near genes with tissue-specific expression are often seen as the evidence that the trait-associated genetic effects are enriched in particular tissues. This implicitly assumes genetic variants with tissue-specific genetic effects on gene expression are co-located with genes with tissue-specific expression.”

“This is an important result and challenges a current dogma that focus interest on GWAS association results in genes that are differentially expressed in the tissue of most relevance to the disease.”

I also didn't understand how the authors reached the conclusion that "Our results reinforce the need to generate tissue-specific eQTL data sets to identify variants that generate variation between people in a specific tissue regardless of the relative expression level of the tissue" -- I thought a major point of this manuscript is that the tissue does not matter too much and we can use e.g. blood instead of brain.

Re: We apologize for the confusion. Our conclusion is that large sample size is a key factor in identifying eQTLs, and that much power can be gained from the use of large samples in a specific tissue. We have rephrased the sentence to (page 11) “Our results reinforce that very large sample sizes are needed to generate eQTL data sets in a specific tissue (e.g. blood) to increase the power of detecting cis-eQTLs regardless of the relative expression level of the tissue.”.

The authors write that Backenroth et al did not study eQTLs, but they did; see e.g. Table 1 of Backenroth et al. Why such different results here? Please discuss.

Re: We apologize for the confusion. In previous version of the response, we stated that “These studies, however, did not look at eQTL effects”. By “these studies” we meant the Heintzman et al. (2009 Nature) and Visel et al. (2009 Nature) studies. Backenroth et al did study eQTLs and concluded that eQTLs identified in a particular tissue tend to be enriched among the predicted functional variants in a relevant Roadmap tissue. We have commented on this in the revised manuscript (page 6) : “Previous studies have indicated that chromatin state at promoters is largely invariant across diverse cell types whereas enhancers are marked with highly cell-type-specific histone modification patterns (Heintzman et al. 2009 Nature), that functional variants (predicted by chromatin activity data) in enhancers are less likely to be shared across tissues compared with those in promoters (Backenroth et al. 2017 bioRxiv), and that cell type-specific eQTLs are more dispersedly distributed around the transcription start site than eQTLs affected expression in multiple cell types (Dimas et al. 2009 Science; Fairfax et al. 2012 Nature Genetics). These results seem to indicate that tissue- specific eQTLs are enriched in distal regulatory elements (i.e. enhancers). To address this hypothesis, we ...”

We also **removed** the following statements from the manuscript to avoid confusion “These results do not support the hypothesis that eQTLs with tissue-specific effects are more likely to be located in enhancers”.

I still find the methods confusingly written -- in particular, what depends on the SNP, what depends on the gene, what depends on the tissue, and what is fixed vs random where. More subscripts might help here, along with an indication of what is varying under "cov" and "var". As an example of what is confusing: it seems like in the first part of the methods, $\text{cov}(\hat{\beta}_i, \hat{\beta}_j)$ is observed, while in the second part of the method, it must be estimated. There should be different notation for different quantities.

Re: We thank the reviewer for this comment. We have re-written the Methods section to clarify notation (pages 15,16, and 17).

We first clarify that there is only one SNP (i.e. the top cis-eQTL) per gene except for that in the analysis to compute the approximated sampling correlation from "null" SNPs. Hence, most variables (e.g. b , \hat{b} and e) vary across genes. We use subscripts i and j to represent two tissues. We use "var" and "cov" to represent the variance of a variable in a tissue and covariance of a variable between two tissues, respectively, and " $\widehat{\text{var}}$ " and " $\widehat{\text{cov}}$ " to represent their estimates in a specific set of genes. We denote " V " and " C " as the sampling variance of an estimand and sampling covariance between two estimands across repeated experiments, respectively, and " \widehat{V} " and " \widehat{C} " as their estimates (e.g. by a Jackknife approach). We have further clarified which parameters are estimated and which are approximated.

We have also made the corresponding changes in the **Supplementary Note 1**.

A minor point: I'm not sure the authors understood my comment about a correlation of 1 not indicating equality. x and $x/2$ have a correlation of 1, even if both are in the same units.

Re: The reviewer is correct that a correlation of 1 does not necessarily indicate equality between x and y unless the variances of x and y are identical.

We have reported the variance of cis-eQTL effects across genes in different brain region (**Supplementary Fig. 21**), and revised the relevant text as (page 9):

"We therefore can approximately estimate n_{eff} of GTEx-brain assuming constant mean q^2 across brain regions (**Supplementary Note 2**). Note that this assumption is justified by the highly consistent estimates of the variance of cis-eQTL effects across genes in different brain regions (**Supplementary Fig. 21**) along with a mean r_b estimate of 0.94 between pairwise brain regions for cis-eQTL effects in SD units."

Reviewer #1 had no further comments to the authors.

Reviewer #3 (Remarks to the Author):

The manuscript is much improved, and the authors have responded adequately to almost all of the points I raised previously. I particularly appreciate the better contextualization of the results and the much clearer methods section.

The main concern I have remaining is still the comparison of difference of eQTL effect size vs difference of expression level. In particular, the authors are excluding some types of correlation but not others. Suppose, for example, that every top eQTL explains exactly 1 SD of the expression of the corresponding gene, in both tissues. In RPKM units, this would lead to a strong correlation between expression level and eQTL effect size, which could support, for example, an eQTL-based explanation of the heritability enrichment near specifically expressed genes in relevant tissues. Similarly, because the authors are only excluding some types of correlations, I don't see how the statement "selection of genes at $P_{\text{(cis-eQTL)}} < 5 \times 10^{-8}$ in muscle for the rb analysis above was not a cause of enrichment (or depletion) of genes with tissue-specific expression" follows from the analysis, at least without a more careful justification. It could be that other ways of doing this analysis would also be problematic, but the authors should still be careful to adequately caveat the analysis they did choose to do, to characterize what types of relationships would and wouldn't be picked up by it, and then to justify how the particular conclusions they come to are supported by the analysis they did.

Two minor comments:

1. I remain convinced that a discussion of Table 1 of Backenroth et al., which shows enrichment of tissue-specific eQTLs in tissue-specific regulatory elements, is important. This is separate from the results of Backenroth et al. that are currently mentioned.
2. In the description of MECS in the methods section, I think you meant for θ to have an i, j subscript. As is, it sounds like you are making a very strong assumption that correlation is constant across pairs of tissues, but I think this is likely a typo.

Response to Reviewers' Comments

Reviewer #1 had no further comments to the authors.

Reviewer #3 (Remarks to the Author):

The manuscript is much improved, and the authors have responded adequately to almost all of the points I raised previously. I particularly appreciate the better contextualization of the results and the much clearer methods section.

Re: We thank the reviewer for the positive remark that our manuscript is much improved.

The main concern I have remaining is still the comparison of difference of eQTL effect size vs difference of expression level. In particular, the authors are excluding some types of correlation but not others. Suppose, for example, that every top eQTL explains exactly 1 SD of the expression of the corresponding gene, in both tissues. In RPKM units, this would lead to a strong correlation between expression level and eQTL effect size, which could support, for example, an eQTL-based explanation of the heritability enrichment near specifically expressed genes in relevant tissues. Similarly, because the authors are only excluding some types of correlations, I don't see how the statement "selection of genes at $P_{\text{cis-eQTL}} < 5 \times 10^{-8}$ in muscle for the r_b analysis above was not a cause of enrichment (or depletion) of genes with tissue-specific expression" follows from the analysis, at least without a more careful justification. It could be that other ways of doing this analysis would also be problematic, but the authors should still be careful to adequately caveat the analysis they did choose to do, to characterize what types of relationships would and wouldn't be picked up by it, and then to justify how the particular conclusions they come to are supported by the analysis they did.

Re: We agree with the reviewer and have revised the text as (page 7): "However, these results also suggest that an eQTL with identical effect on gene expression in SD units in different tissues could show different effects in RPKM units if the variance of gene expression varies across tissues, which might explain the results from recent studies that genetic variants in or near genes differentially expressed in a particular tissue are enriched for associations with a complex trait."

We also agree with the reviewer that the statement below is not fully justified and thus have removed it from the manuscript.

"The lack of correlation between test-statistic for difference in SNP effect on gene expression and test-statistic for difference in expression level of the corresponding gene also means that the selection of genes at $P_{\text{cis-eQTL}} < 5 \times 10^{-8}$ in muscle for the r_b analysis above was not a cause of enrichment (or depletion) of genes with tissue-specific expression."

Two minor comments:

1. I remain convinced that a discussion of Table 1 of Backenroth et al., which shows enrichment of tissue-specific eQTLs in tissue-specific regulatory elements, is important. This is separate from the results of Backenroth et al. that are currently mentioned.

Re: We thank the referee for this comment. We have commented on the result in Table 1 of Backenroth et al. (2017 bioRxiv) in the revised manuscript (pages 6 and 7).

"Note that these results do not contradict the observation from a recent study that eQTLs detected in specific tissues in GTEx tend to be most enriched among the variants predicted to be functional in relevant REMC tissues (Backenroth et al. 2017 bioRxiv)."

2. In the description of MECS in the methods section, I think you meant for θ to have an i,j subscript. As is, it sounds like you are making a very strong assumption that correlation is constant across pairs of tissues, but I think this is likely a typo.

Re: Yes, it is a typo. We have fixed that in the revised manuscript (page 17).